# Dislocation Density of Electron Beam Powder Bed Fusion Ti−6Al−4V Alloys Determined via Time-Of-Flight Neutron Diffraction Line-Profile Analysis

Kenta Yamanaka [1,*], Manami Mori [1,2], Yusuke Onuki [3,†], Shigeo Sato [4] and Akihiko Chiba [1]

[1] Institute for Materials Research, Tohoku University, 2-1-1 Katahira, Aoba-ku, Sendai 980-8577, Japan
[2] Department of General Engineering, National Institute of Technology, Sendai College, 48 Nodayama, Medeshima-Shiote, Natori 981-1239, Japan
[3] Frontier Research Center for Applied Atomic Sciences, Ibaraki University, 162-1 Shirakata, Tokai 319-1106, Japan
[4] Graduate School of Science and Engineering, Ibaraki University, 4-12-1 Nakanarusawa, Hitachi 316-8511, Japan
* Correspondence: kenta.yamanaka.c5@tohoku.ac.jp
† Current address: Department of Advanced Machinery Engineering, School of Engineering, Tokyo Denki University, 5 Senju Asahi-cho, Adachi-ku, Tokyo 120-8551, Japan.

**Abstract:** Ti–6Al–4V alloys undergo a multiple phase transformation sequence during electron beam powder bed fusion (EB-PBF) additive manufacturing, forming unique dislocation substructures. Thus, determining the dislocation density is crucial for comprehensively understanding the strengthening mechanisms and deformation behavior. This study performed time-of-flight neutron diffraction (TOF-ND) measurements of Ti–6Al–4V alloys prepared via EB-PBF and examined the dislocation density in the as-built and post-processed states using convolutional multiple whole profile (CMWP) fitting. The present TOF-ND/CMWP approach successfully determined the bulk-averaged dislocation density $(6.8 \times 10^{13}\ \mathrm{m}^{-2})$ in the as-built state for the $\alpha$-matrix, suggesting a non-negligible contribution of dislocation hardening. The obtained dislocation density values were comparable to those obtained by conventional and synchrotron X-ray diffraction (XRD) measurements, confirming the reliability of the analysis, and indicating that the dislocations in the $\alpha$-matrix were homogeneously distributed throughout the as-built specimen. However, the negative and positive neutron scattering lengths of Ti and Al, respectively, lowered the diffraction intensity for the Ti−6Al−4V alloys, thereby decreasing the lower limit of the measurable dislocation density and making the analysis difficult.

**Keywords:** additive manufacturing; electron beam powder bed fusion (EB-PBF); Ti−6Al−4V alloy; duplex microstructures; hot isostatic pressing (HIP); time-of-flight neutron diffraction (TOF-ND); convolutional multiple whole profile (CMWP); dislocation density





## 1. Introduction

Additive manufacturing (AM) technologies, which produce three-dimensional (3D) objects in a layer-by-layer manner, offer distinct advantages including part/production flexibility, efficient material and energy usage, and reduced lead times [1–3]. The major target areas include the aerospace, medical, dental, energy, and transportation industries. Although a variety of metal AM processes have been proposed [1,4], the electron beam powder bed fusion (EB-PBF) process [5,6] offers particular benefits, such as negligible residual stress and distortion of built parts. Moreover, a wide range of process parameters, such as the power and scanning speed of the electron beam, and a unique preheating procedure [7,8], are beneficial for minimizing building defects. Therefore, EB-PBF has been applied to a variety of metals and alloys, such as titanium [9–16], biomedical Co–Cr–Mo alloys [17–19], Ni-based superalloys [20–25], steels [26–29], copper [30,31], and high-entropy alloys [32–36].

In the EB-PBF process, a high-energy electron beam scans a metal powder bed, creating a highly localized melt pool that enables rapid heating and cooling [4,5]. Accordingly, during PBF fabrication, Ti–6Al–4V alloy, a titanium alloy typically used for various applications in the aerospace and biomedical industries [37–40], experiences a martensitic transformation from the body-centered cubic (bcc) β-phase to hexagonal close-packed (hcp) α'-martensite [11,13,41,42], which is generally unachievable in conventional manufacturing. The fully α'-martensitic microstructure that is obtained in laser beam PBF (LB-PBF) generally exhibits high strength while sacrificing its ductility [13,41]. Furthermore, in the EB-PBF process, α'-martensite is held for several hours at an elevated temperature that is similar to the preheating temperature, leading to an equilibrium α + β microstructure [11]. Thus, the decomposition behavior of α'-martensite is crucial for understanding the microstructural evolution and ultimately optimizing its mechanical properties.

The yield strength of the EB-PBF Ti–6Al–4V alloys with acicular α + β microstructures has been generally modeled using the α-lath width, based on the traditional Hall–Petch relationship [12,14]. However, multiple strengthening mechanisms are operative in metals and alloys. Among the various factors involved in the strengthening of metallic materials, it is generally challenging to determine the dislocation density quantitatively. The evolution of dislocation substructures has been reported in various AM materials, particularly in LB-PBF [43–45]. In a previous study by the current authors, synchrotron X-ray diffraction (XRD) line-profile analysis was performed to examine the dislocation densities of EB- and LB-PBF-processed Ti–6Al–4V alloys [46]. As expected, a high dislocation density ($4.3 \times 10^{15}$ m$^{-2}$) was obtained for a fully martensitic LB-PBF Ti–6Al–4V alloy, significantly contributing to the high strength that is generally observed in LB-PBF Ti–6Al–4V alloys. Although a part of dislocations in the initially formed α'-martensite was diminished during the high-temperature EB-PBF process, the dislocation density ($6.1 \times 10^{13}$ m$^{-2}$) of the α-matrix of the as-built EB-PBF specimen and its contribution (105–169 MPa) to the overall strengthening mechanisms were not negligible. Notably, high-energy synchrotron radiation enabled the simultaneous determination of the dislocation density for the α-matrix and nanosized β-phase, and an enhanced dislocation accumulation in the β-phase was discovered. These results highlight the significance of multiple phase transformations during in-process high-temperature exposures.

Conversely, compared with the other characterization techniques, neutron diffraction (ND) obtains bulk-averaged microstructural characteristics from a relatively large volume ($\sim$10 cm$^3$) and helps establish a correlation with the macroscopic mechanical properties and plastic deformation behavior. The ND technique was initially employed in the field of metal AM to evaluate residual stress [47–49]. A pioneering study by Wu et al. [47] systematically examined the laser scanning strategy (e.g., pattern, power, and speed) on the residual stress of LB-PBF 316L stainless steels and indicated directions toward mitigating residual stress. Wang et al. [48] emphasized that a location-dependent variation in composition, which originates from elemental volatilization during the AM process, could alter the reference lattice spacing, generating experimental errors. Further, the application of ND to microstructural/mechanical behavior has been attracting attention [50–52]. Ghorbanpour et al. [52] studied the stress−strain response of additively manufactured Inconel 718 at room and elevated temperatures. The ND-based texture analysis was used to calibrate and validate the developed elasto-plastic crystal plasticity model. Recently, by combining time-of-flight ND (TOF-ND) measurements and Rietveld texture analysis (RTA) [53], the β-phase fraction in an EB-PBF Ti−6Al−4V alloy was precisely determined [54]. Furthermore, a hierarchal texture evolution between the prior β-phase, acicular α-matrix, and β-phase at the α-interfaces was observed. However, the application of ND microstructural analyses for additively manufactured Ti−6Al−4V alloys is limited [55–59], indicating the need to examine the potential feasibility of ND-based line-profile analysis.

In this study, TOF-ND measurements were performed on Ti−6Al−4V alloy specimens prepared via EB-PBF at J-PARC, Japan, and the dislocation density using line-profile analysis was evaluated. The results were compared with those obtained using conventional

and synchrotron XRD measurements. A direct comparison of dislocation analysis based on different techniques, such as XRD and ND, has rarely been reported; therefore, this study can be highly beneficial for the material characterization community.

## 2. Materials and Methods

### 2.1. Sample Preparation

Ti−6Al−4V alloy samples that were employed in this study are similar to those used in previous studies [46,54,60]. Cylindrical specimens with a diameter of φ18 mm and height of 160 mm, where the longitudinal directions of the specimens were parallel to the building directions (BDs), were prepared using an Arcam A2X system with the standard raw powder and build parameters (Table 1) provided by the manufacturer. The specimens were almost fully dense, with a representative relative density of 99.3%, as determined using Archimedes' principle. Table 2 lists the chemical composition of the as-built specimens.

**Table 1.** Building parameters used for sample preparation.

| Parameter | Value |
|---|---|
| Powder size (μm) | 45–100 |
| Power (W) | 240–1260 |
| Scan speed (mm s$^{-1}$) | ~500 |
| Layer thickness (μm) | 50 |
| Line offset (mm) | 0.1 |
| Focus offset (mA) | 3 |
| Dwell time (s) | 20 |
| Preheating temperature (°C) | 730 |

**Table 2.** Chemical composition of the as-built specimen used in this study (mass%).

| | Ti | Al | V | Fe | O | N | H |
|---|---|---|---|---|---|---|---|
| As-built | Bal. | 6.17 | 4.01 | 0.20 | 0.11 | 0.014 | 0.003 |
| ASTM Grade 5 | Bal. | 5.50−6.75 | 3.50−4.50 | ≤0.30 | ≤0.20 | ≤0.05 | ≤0.015 |

To examine the effect of post-processing, hot isostatic processing (HIP) was performed on the as-built specimens at 920 °C for 2 h under an argon gas pressure of 100 MPa. Additionally, conventional heat treatment (HT) at 920 °C for 2 h was conducted in a muffle furnace without applying atmospheric pressure. Hereafter, these specimens are referred to as HIP and HT specimens. Both specimens were furnace-cooled in each piece of equipment after the HT. The mechanical properties of the investigated specimens were summarized in a previous study [60].

### 2.2. Electron Microscopy

Scanning electron microscopy (SEM) observations of the prepared specimens were performed using a JSM-IT800 instrument (JEOL, Tokyo, Japan) operating at 15 kV. Cross-sections parallel to the BD were prepared by grinding with emery papers and finished with a mixture of colloidal silica suspension (OP-S, Struers, Tokyo, Japan), $H_2O_2$ solution, and distilled water. Scanning transmission electron microscopy (STEM) observations and energy-dispersive X-ray spectroscopy (EDS) mapping were performed on a dual spherical aberration-corrected TITAN$^3$ G2 60–300 S/TEM (FEI, Hillsboro, OR, USA) operating at 300 kV. A focused ion beam (FIB) process (Versa 3D Dual Beam, FEI, Hillsboro, OR, USA) was employed for sample preparation for the STEM-EDS analysis.

### 2.3. Neutron Diffraction Measurements

TOF-ND measurements of the specimens were conducted at 500 kW using an iMATE-RIA (BL20), J-PARC, Japan [61]. Samples with φ9.8 mm in diameter and a height of 30 mm

were cut from the specimens. Each sample was fixed to a goniometer using a vanadium tube ($\varphi$10 mm inner diameter), which was insensitive to neutron beams, during the TOF-ND measurements. The acquiring time was ~30 min for each measurement. The iMATERIA beamline consists of three types of detector banks: backscattering (BS), special environment (SE), and low angle (LA), covering a wide range of scattering vectors. The ND datasets used in this study were obtained simultaneously with those used in the previous study [54].

### 2.4. Line-Profile Analysis

The dislocation density was determined by the convolutional multiple whole profile (CMWP) method [62,63] using the diffractogram obtained from the entire BS bank of the iMATERIA. The measured profiles were fitted using convolution profile $I$ as follows:

$$I = I_s \otimes I_m \otimes I_i \tag{1}$$

where $I_s$, $I_m$, and $I_i$ are functions of the crystallite size, microstrain, and instrumental broadening, respectively. $I_i$ was determined from standard $LaB_6$ powder (SRM-660b, NIST). $I_m$ is given as [64,65]:

$$I_m = \exp\left(-2\pi^2 g^2 L^2 \langle \varepsilon_{g,L}^2 \rangle\right) \tag{2}$$

where $g$ is the absolute value of the diffraction vector, and $L$ is the Fourier variable. The dislocation density ($\rho$) was deduced from the mean square strain ($\langle \varepsilon_{g,L}^2 \rangle$), as follows [64]:

$$\langle \varepsilon_{g,L}^2 \rangle = \left(\rho \overline{C} b^2 / 4\pi\right) f(L/R_e) \tag{3}$$

where $\overline{C}$, $b$, $f$, and $R_e$ are the average contrast factors of the dislocations, absolute value of the Burgers vector, Wilkens function, and effective outer cutoff radius of the dislocations, respectively [64–66]. The function $f(L/R_e)$ was determined for the distribution of dislocations over the entire $L$ range, from zero to infinity [64]. Diffractograms for CMWP analysis require a high angle resolution; thus, datasets from the BS bank were employed.

## 3. Results

### 3.1. Microstructures

Figure 1 shows SEM backscatter electron (BSE) images of the specimens prepared in this study. The as-built specimen (Figure 1a,b) had a fine acicular $\alpha + \beta$ microstructure, where the $\alpha$-lath width was determined to be $1.7 \pm 0.2$ μm [60]. The nanosized $\beta$-phase, which was enriched with vanadium, exhibits a higher contrast at the $\alpha$-lath interfaces. The post-processed specimens exhibited coarser acicular microstructures while maintaining $\alpha + \beta$ microstructures. A previous study [54] revealed that the volume fractions of the $\beta$-phase in the as-built, HIP, and HT specimens were 4.9, 7.4, and 9.0 vol.%, respectively, as determined via TOF-ND/RTA.

Figure 2a shows a bright-field (BF)-TEM image of the as-built specimen. $\alpha$-Laths with interfacial nanosized $\beta$-phase precipitates are observed. In the high-angle annular dark-field (HAADF)-STEM image in Figure 2b, the higher contrast in the HAADF-STEM image demonstrates the nanoscale $\beta$-phase precipitation that occurred at the $\alpha$-lath interfaces. The corresponding STEM-EDS mappings in Figure 2c–e represent the V-rich $\beta$-phase, whereas Ti and Al, as $\alpha$-stabilizing elements, are preferentially distributed in the $\alpha$-matrix. Figure 2f shows a high-magnification annular bright-field (ABF)-STEM image representing the vicinity of the $\beta$-phase precipitate. Dislocations are observed in both the $\alpha$-matrix and the $\beta$-phase. Notably, a significant strain contrast is observed in the $\alpha$-matrix surrounding the $\beta$-phase precipitates, suggesting that dislocations are introduced via precipitation of the $\beta$-phase. However, it is difficult to quantify the dislocation density using STEM images.

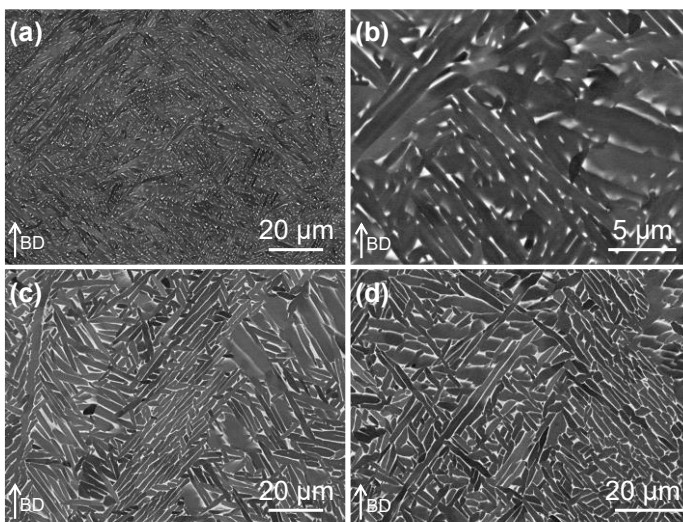

**Figure 1.** SEM-BSE images of the (**a**,**b**) as-built, (**c**) HIP, and (**d**) HT specimens.

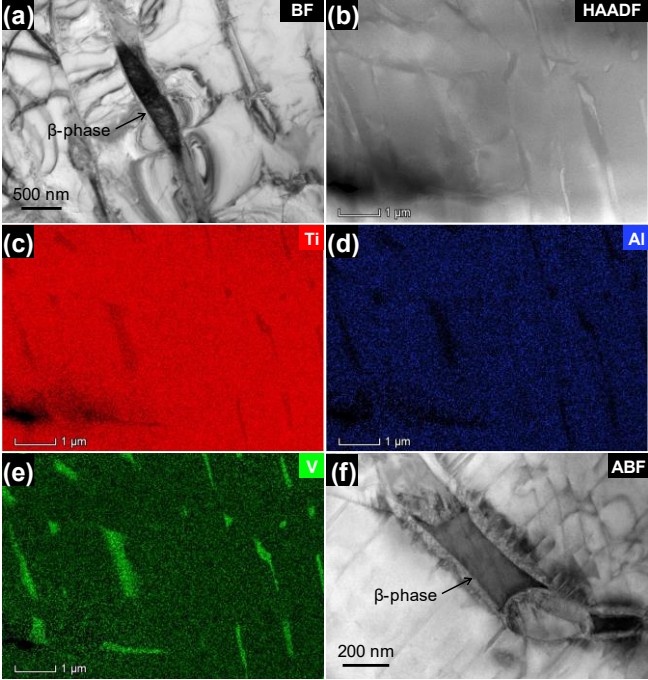

**Figure 2.** (**a**) BF-TEM image, (**b**) HAADF-STEM image, and corresponding STEM-EDS maps of (**c**) Ti, (**d**) Al, and (**e**) V for the as-built specimen. (**f**) Magnified ABF-STEM image showing the vicinity of the β-phase precipitate.

### 3.2. Neutron Diffraction Patterns

Figure 3 shows the TOF-ND patterns for the as-built, HIP, and HT specimens obtained from the BS bank of the iMATERIA. Multiple reflections are successfully detected for each constituent phase; thus, the simultaneous determination of the dislocation density is possible for both the α- and β-phases. Neither post-processing method changes the phase constituents (i.e., α- and β-phases), although the β-phase fraction increases [54].

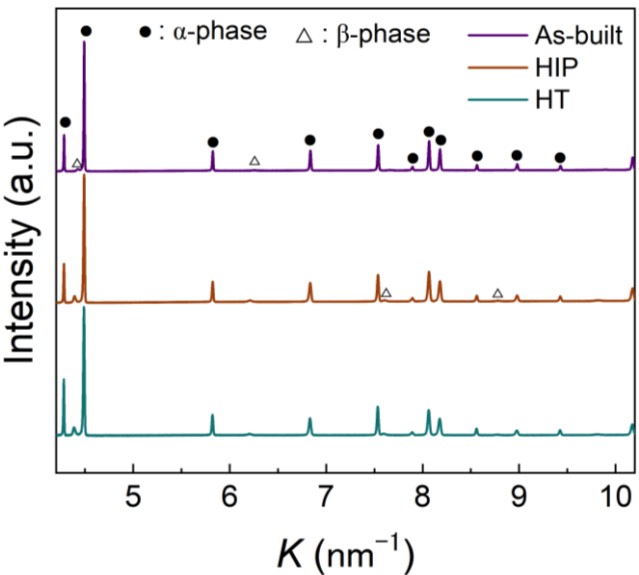

**Figure 3.** TOF-ND profiles for the as-built, HIP, and HT specimens, as obtained at the BS bank of the iMATERIA beamline at J-PARC.

### 3.3. Dislocation Density

Figure 4 shows the result of the CMWP fitting for the diffractogram of the as-built specimen. Both α- and β-phases are successfully fitted using the CMWP method. In contrast, because of the lower peak broadening caused by the decreased dislocation density, it was not possible to evaluate the dislocation densities of both post-processed specimens by TOF-ND-based CMWP analysis.

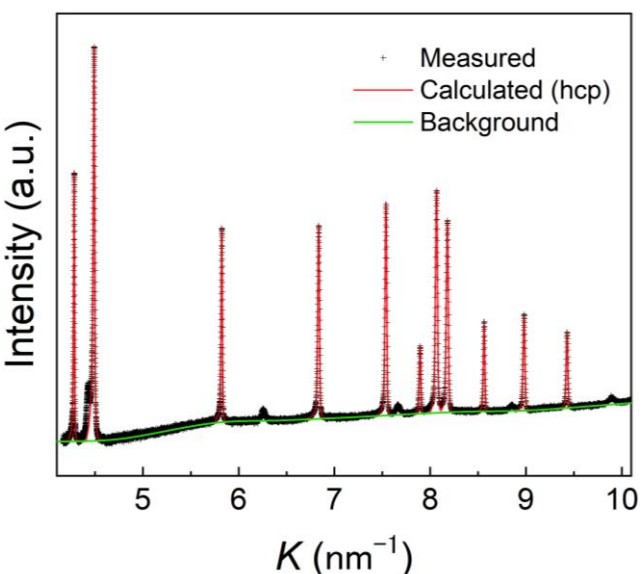

**Figure 4.** An example of the fitting result for the TOF-ND pattern of the as-built specimen using the CMWP method.

The results obtained from the CMWP fitting are listed in Table 3. For comparison, the results obtained for the identical as-built specimens using conventional and synchrotron XRD measurements [46] are also presented. The obtained dislocation density for the α-matrix of the as-built specimen is $6.8 \times 10^{13}$ m$^{-2}$, which is consistent with those obtained using the conventional and synchrotron XRD results for the same as-built specimen. Notably, the dislocation density of the nanosized β-phase could not be determined, although multiple β-peaks that are suitable for the CMWP analysis are detected. Furthermore, the

CMWP method can determine the crystallite size, which can be correlated with the size of the dislocation arrays, subgrains, and cell structures [62], and the dislocation arrangement parameter, which indicates the interactions between dislocations [64]. However, the relatively low dislocation density in the order of $10^{13}$ m$^{-2}$ increases the crystallite size to several micrometers, resulting in negligible size effects. Therefore, these two parameters are not shown.

**Table 3.** Dislocation density of the as-built Ti−6Al−4V specimen, as determined via TOF-ND measurements and CMWP method. For comparison, those obtained from conventional and synchrotron XRD measurements [46] are also listed.

| Diffraction Source | Phase | Dislocation Density (m$^{-2}$) |
|---|---|---|
| Neutron (this study) | α | $(6.8 \pm 0.7) \times 10^{13}$ |
| | β | − |
| Synchrotron radiation [46] | α | $(6.1 \pm 0.2) \times 10^{13}$ |
| | β | $(7.7 \pm 2.0) \times 10^{14}$ |
| Conventional X-ray [46] | α | $(7.5 \pm 0.5) \times 10^{13}$ |
| | β | − |

## 4. Discussion

The dislocation densities of the as-built specimens determined from the TOF-ND data are consistent with those obtained using conventional X-ray and synchrotron radiation (Table 3). Thus, the current technique enables the reliable quantification of the as-built dislocation structure. Notably, the current ND measurements captured considerably larger sample volumes ($\varphi$9.8 mm $\times$ 30 mm) than those in the previous synchrotron XRD study (a spot size of ~500 $\times$ 200 $\mu$m$^2$ and a sample thickness of ~1 mm) [46]. Thus, the similar dislocation density values obtained in both techniques imply that dislocations were homogeneously distributed throughout the as-built specimen. The dislocation density of the α-matrix ($6.8 \times 10^{13}$ m$^{-2}$) is not significantly high but cannot be considered negligible when modeling the strengthening mechanisms [46]. The dislocation substructure in the α-matrix of the as-built specimen can be caused by the residual dislocations introduced via the β → α′ martensitic transformation. Moreover, Figure 2f suggests that the volume expansion for the α′-to-β phase transformation could be an additional dislocation source in the α-matrix, as visible strain contrasts are identified near the nanosized β-phase precipitates. Therefore, dislocation hardening evidently occurs in the as-built EB-PBF specimen. By contrast, the dislocation density after heat treatment (i.e., HIP and HT samples) was below the lower detection limit of the present TOF-ND line-profile analysis, even though a conventionally prepared Ti–6Al–4V alloy specimen after annealing at ~700 °C contained numerous dislocations [46]. Because AM Ti–6Al–4V alloys are often subjected to HIP, this is also important in terms of strength.

The overall strengthening of duplex α + β titanium alloys is described as follows [67]:

$$\Delta\sigma = \Delta\sigma_\alpha \, (1 - f_\beta) + \Delta\sigma_\beta \, f_\beta \tag{4}$$

where $\Delta\sigma_\alpha$ and $\Delta\sigma_\beta$ are the α- and β-phase strengths, respectively, and $f_\beta$ is the volume fraction of the β-phase. The strengthening mechanisms that should be considered include dislocation hardening for each phase, as described by the Bailey–Hirsch (or Taylor) relationship [66]:

$$\Delta\sigma_D = \alpha M_T G b \rho^{1/2} \tag{5}$$

where $\Delta\sigma_D$ represents the increment in the yield strength owing to dislocation strengthening, and $\alpha$, $M_T$, $G$, and $b$ are the dislocation interaction term, Taylor factor, shear modulus, and magnitude of the Burgers vector, respectively. The crystallographic orientation distribution also affects the Taylor factor in the modeling of the strengthening mechanisms. In a previous study [54], the phase fraction and crystallographic textures were precisely

determined using Rietveld texture analysis [53] of TOF-ND data. Thus, the TOF-ND measurements at the iMATERIA, which can simultaneously evaluate the dislocation density, phase fraction, and texture, can potentially facilitate the comprehensive understanding of the strengthening mechanisms of EB-PBF Ti−6Al−4V alloys. Notably, compared with synchrotron XRD, the high penetration ability of the neutron source is beneficial for bulk-averaged dislocation density determination, which is essential for establishing the relationship between the microstructure and mechanical properties. However, determining the dislocation density of the interfacial nanosized β-phase in the EB-PBF Ti−6Al−4V alloys is challenging. The electron backscattering diffraction (EBSD) technique has previously been employed to measure the distributions of geometrically necessary dislocations [68]. However, a certain step size (generally larger than ~30 nm) in the EBSD measurements is not always sufficient to analyze the β-phase precipitates obtained in EB-PBF. Moreover, conventional XRD measurements cannot detect multiple β-peaks that are suitable for the CMWP analysis. Notably, in a previous study by the current authors [46], synchrotron XRD line-profile analysis successfully determined the dislocation density in the as-built state for the α-matrix and interfacial nanosized β-phase simultaneously. Furthermore, enhanced dislocation substructures in the nanosized β-phase due to plastic accommodation were observed. However, unlike the synchrotron XRD results [46], the dislocation density of fine β-phase precipitates in the acicular duplex microstructure could not be determined by the current ND measurements at J-PARC. For Ti−6Al−4V alloys, the neutron scattering length values for Ti and Al are negative and positive, respectively, resulting in a lower ND intensity. Thus, compared with the synchrotron XRD results [46], the accuracy of the line-profile analysis of the Ti−6Al−4V alloys is potentially low, depending on the measurement conditions. Thus, the optimization of the measurement conditions is critical. Notably, increasing the beam power (up to 1 MW at J-PARC) is crucial for analyzing the dislocation density of the β-phase.

## 5. Conclusions

In this study, TOF-ND measurements of as-built and post-processed EB-PBF Ti–6Al–4V alloy specimens were performed to evaluate the dislocation density quantitatively using the CMWP method. The obtained dislocation densities were compared with those determined using conventional and synchrotron XRD measurements for the same specimen. The findings of this study are summarized as follows:

(1) The as-built specimen exhibited a fine acicular α + β microstructure, attributable to the β → α′ martensitic transformation and subsequent decomposition of the α′ martensite during the EB-PBF process.

(2) The TOF-ND/CMWP approach that was utilized successfully determined the bulk-averaged dislocation density for the α-matrix, revealing a non-negligible contribution of dislocation hardening in the as-built specimen.

(3) The obtained dislocation density values were comparable to those obtained using conventional and synchrotron XRD measurements, suggesting that CMWP fitting for the same ND data provided a reliable dislocation density.

(4) The insignificant difference in dislocation density between XRD and ND suggested that the dislocations that evolved during EB-PBF were homogeneously distributed throughout the as-built specimen.

(5) The negative and positive neutron scattering lengths of Ti and Al, respectively, lowered the diffraction intensity of the Ti−6Al−4V alloys, potentially reducing the accuracy of the analysis. The limitation of ND, as opposed to synchrotron XRD, is the difficulty in dislocation analysis of nanoscale β-phase precipitates.

**Author Contributions:** Conceptualization, K.Y.; methodology, Y.O. and S.S.; validation, K.Y. and S.S.; investigation, K.Y., M.M., Y.O. and S.S.; resources, A.C.; data curation, K.Y. and M.M.; writing—original draft preparation, K.Y.; writing—review and editing, M.M., Y.O. and S.S.; visualization,

K.Y. and M.M.; supervision, S.S. and A.C.; project administration, K.Y.; funding acquisition, K.Y. All authors have read and agreed to the published version of the manuscript.

**Funding:** This research was financially supported by a Grant-in-Aid for Transformative Research Areas (A), the Japan Society for the Promotion of Sciences (JSPS) (Grant No. 22H05274), the Iketani Science and Technology Foundation, Japan (Grant No. 0291050-A), the Japan Titanium Society, and the Light Metal Educational Foundation, Inc., Japan.

**Data Availability Statement:** The data presented in this study are available on request from the corresponding author.

**Acknowledgments:** The authors would like to thank Xiaoli Shui, Kenya Kurita, and Yoshihiko Nagata for sample preparation and Yumiko Kodama and Yuichiro Hayasaka for FIB sampling and TEM observations. Neutron diffraction experiments were performed at J-PARC MLF, Japan, under user programs (proposal No. 2020PM2007).

**Conflicts of Interest:** The authors declare no conflict of interest.

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
