# Peer review of "Dislocation Density of Electron Beam Powder Bed Fusion Ti–6Al–4V Alloys Determined via Time-Of-Flight Neutron Diffraction Line-Profile Analysis"

_metals, doi:10.3390/met13010086_

Round 1

Reviewer 1 Report

This study showed a method for dislocation density of EB-PBF Ti6Al4V alloy using time-of-flight neutron diffraction line-profile analysis. It is a useful method but is common in the conventional testings. This method is not specially developed for the additive manufacturing. Therefore, in the whole paper, it just displayed a method alone. The author should add some experiments and expand the discussion about the dislocation density that are closely related to the additive manufacturing process itself. Such as, the formation and reduced mechanism of dislocation density, the difference in mechanical properties between samples with different dislocation density. In addition, please pay attention to the repeatability and accuracy of the testing.

Author Response

Comment: This study showed a method for dislocation density of EB-PBF Ti6Al4V alloy using time-of-flight neutron diffraction line-profile analysis. It is a useful method but is common in the conventional testings. This method is not specially developed for the additive manufacturing. Therefore, in the whole paper, it just displayed a method alone. The author should add some experiments and expand the discussion about the dislocation density that are closely related to the additive manufacturing process itself. Such as, the formation and reduced mechanism of dislocation density, the difference in mechanical properties between samples with different dislocation density. In addition, please pay attention to the repeatability and accuracy of the testing.

Reply: Thank you for reviewing our manuscript and providing the valuable comments. As the reviewer pointed out, line-profile analysis employed in this work has not been developed for AM materials. However, as described in the introduction, evaluating dislocation measurements is important for comprehensive understanding of the strengthening mechanisms. Notably, the current ND-based line-profile analysis has been rarely reported.

On the other hand, the similar dislocation density values were obtained in the synchrotron and neutron diffraction techniques. This implies the homogeneity in dislocation density distribution as these two techniques cover different measurable volumes to each other. This discussion has been included in the revised manuscript.

Moreover, we evaluated the as-built and heat-treated specimens (HIP and HT) and found negligible dislocation density after the heat treatment. Our previous study detected some extents of dislocations in a conventionally prepared Ti–6Al–4V bar after annealing at 700 °C. AM Ti–6Al– 4V alloys are often subjected to HIP; thus, this information is also valuable for the consideration in their strength.

Reviewer 2 Report

This paper presents a study of TOF-ND measurements of as-built and post-processed EB-PBF Ti–6Al– 4V alloy specimens were performed to quantitatively evaluate dislocation density using the CMWP method. The research topic undertaken in the article is interesting and original. The study meets scientific standards. Presented results are appropriate and conclusions are interactive with the content of the paper. The references are current and closely related to the topic.

Author Response

Comment: This paper presents a study of TOF-ND measurements of as-built and post-processed EB-PBF Ti–6Al– 4V alloy specimens were performed to quantitatively evaluate dislocation density using the CMWP method. The research topic undertaken in the article is interesting and original. The study meets scientific standards. Presented results are appropriate and conclusions are interactive with the content of the paper. The references are current and closely related to the topic.

Reply: Thank you for reviewing our manuscript and providing the positive feedback. We are happy that the reviewer satisfied with our manuscript.

Reviewer 3 Report

Manuscript numbered “metals-2081624” has been reviewed:

The introduction needs some improvements.

It is suggested to compare the results with other additive manufacturing processes Ti6Al4V alloy.

It is suggested to study the effects of heat treatment temperature.

The following papers are suggested for the introduction section:

Study of anisotropy through microscopy, internal friction and electrical resistivity measurements of Ti-6Al-4V samples fabricated by selective laser melting

Investigation of selective laser melting process for Cu-5Sn alloy on surface roughness, microstructure and mechanical property

Modelling of laser powder bed fusion process and analysing the effective parameters on surface characteristics of Ti-6Al-4V

Author Response

Comment: Manuscript numbered “metals-2081624” has been reviewed:

Reply: Thank you for reviewing our manuscript and providing valuable comments.

Comment: The introduction needs some improvements.

Reply: According to the reviewer’s feedback, we have updated the introduction; some detailed description on the previous ND studies for AM materials have been added.

Comment: It is suggested to compare the results with other additive manufacturing processes Ti6Al4V alloy.

Reply: Thank you for your suggestion. Unfortunately, due to the beam time assignment, we cannot do any additional neutron diffraction analysis at this time. Moreover, the other AM processes are not currently available in our laboratory. Please note that in our previous study published in Additive Manufacturing (Ref. 43), we compared the dislocation density between the EB- and LB-PBF processed Ti−6Al−4V alloys.

Comment: It is suggested to study the effects of heat treatment temperature.

Reply: In this study, we examined the dislocation density of EB-PBF Ti−6Al−4V alloys in the as-built and two heat treated conditions. The results indicated that dislocation density after the heat treatment is negligible or under the detection limit. This has been mentioned in the manuscript. We are planning to study the dislocation density for the different heat treatment conditions in the future beamtime and will report the results in a follow-up paper.

Comment: The following papers are suggested for the introduction section:

Study of anisotropy through microscopy, internal friction and electrical resistivity measurements of Ti-6Al-4V samples fabricated by selective laser melting

Investigation of selective laser melting process for Cu-5Sn alloy on surface roughness, microstructure and mechanical property

Modelling of laser powder bed fusion process and analysing the effective parameters on surface characteristics of Ti-6Al-4V

Reply: Thank you for your suggestion. However, to be honest, the topics of the suggested papers are not closely related to the current work. Particularly, the two papers published in Rapid Prototyping Journal are not available in our institution and studies on the different AM process (SLM) and different material (Cu-5Sn). We will consider these papers in our future papers as references.

Round 2

Reviewer 1 Report

Thanks for the response to my comments. This work is useful for additive manufacturing. However, it will be better if the author can add some experiments and expand the discussion about the dislocation density that are closely related to the additive manufacturing process itself. Such as, the formation and reduced mechanism of dislocation density, the difference in mechanical properties between samples with different dislocation density. This is a suggestion.

Reviewer 3 Report

The article is in acceptable form for publication.